# CD1d Selectively Down Regulates the Expression of the Oxidized Phospholipid-Specific E06 IgM Natural Antibody in *Ldlr^−/−^* Mice

**DOI:** 10.3390/antib9030030

**Published:** 2020-07-03

**Authors:** Tapan K. Biswas, Paul A. VanderLaan, Xuchu Que, Ayelet Gonen, Paulette Krishack, Christoph J. Binder, Joseph L. Witztum, Godfrey S. Getz, Catherine A. Reardon

**Affiliations:** 1Department of Pathology, University of Chicago, Chicago, IL 60637, USA; tapank.biswas@yahoo.com (T.K.B.); pvanderl@bidmc.harvard.edu (P.A.V.); pkrishack@uchicago.edu (P.K.); 2Beth Israel Deaconess Medical Center, Boston, MA 02215, USA; 3Department of Medicine, University of California, San Diego, La Jolla, CA 92093, USA; xque@health.ucsd.edu (X.Q.); agonen@health.ucsd.edu (A.G.); jwitztum@health.ucsd.edu (J.L.W.); 4Department of Laboratory Medicine, Medical University of Vienna, Vienna 1090, Austria; christoph.binder@meduniwien.ac.at

**Keywords:** natural antibodies, CD1d, oxidized lipids, innate immunity

## Abstract

Natural antibodies (NAbs) are important regulators of tissue homeostasis and inflammation and are thought to have diverse protective roles in a variety of pathological states. E06 is a T15 idiotype IgM NAb exclusively produced by B-1 cells, which recognizes the phosphocholine (PC) head group in oxidized phospholipids on the surface of apoptotic cells and in oxidized LDL (OxLDL), and the PC present on the cell wall of *Streptococcus pneumoniae*. Here we report that titers of the E06 NAb are selectively increased several-fold in *Cd1d*-deficient mice, whereas total IgM and IgM antibodies recognizing other oxidation specific epitopes such as in malondialdehyde-modified LDL (MDA-LDL) and OxLDL were not increased. The high titers of E06 in *Cd1d*-deficient mice are not due to a global increase in IgM-secreting B-1 cells, but they are specifically due to an expansion of E06-secreting splenic B-1 cells. Thus, CD1d-mediated regulation appeared to be suppressive in nature and specific for E06 IgM-secreting cells. The CD1d-mediated regulation of the E06 NAb generation is a novel mechanism that regulates the production of this specific oxidation epitope recognizing NAb.

## 1. Introduction

Natural antibodies (NAbs) are antibodies with germ-line or near germ-line encoded variable regions [1,2,3] that have broad specificity for foreign and self-antigens. They are mainly produced by a small subset of long-lived, self-replenishing B-1 cells [4,5,6]. IgM isotype NAbs are important regulators of tissue and immune homeostasis and form the first line of defense against microbial infections of viral and bacterial origin [1,2,3]. Many NAbs (>30%) recognize oxidation-specific epitopes (OSE) that form in the presence of oxidative stress [7,8,9]. One important function of NAbs is the recognition of apoptotic cells, in part through the recognition of OSE on their cell membrane [10,11]. This not only promotes the efficient clearance of the apoptotic cells by phagocytes (efferocytosis) but also dampens the inflammatory response by removing cell debris that can promote inflammatory and autoimmune diseases. By targeting OSE, as well as other epitopes, these antibodies have diverse functions related to the prevention of autoimmunity, B cell homeostasis and immune regulation, and they are thought to have a protective role in a variety of pathophysiologies such as atherosclerosis, rheumatoid arthritis, systemic lupus erythematosus, Alzheimer’s disease, multiple sclerosis and perhaps cancer.

Our interest in NAbs is related to their role in atherosclerosis. It is now well known that oxidized LDL (OxLDL), which plays a central role in promoting atherosclerosis, carries OSE [12]. There are several known innate NAbs that recognize OSE on OxLDL [8] and the titers of these antibodies are increased in atherosclerosis-prone hyperlipidemic mice and negatively correlated with risk of cardiovascular disease in humans [13,14,15]. Considerable evidence supports an atheroprotective role of IgM NAbs and, in particular, those specific for OSE [16,17,18,19,20]. The E06 IgM NAb [21] recognizes the phosphocholine (PC) headgroup in oxidized phospholipid (OxPL), but does not recognize PC in native, non-oxidized PC-containing phospholipid. E06 recognizes OxPL present on OxLDL but not native LDL, and on apoptotic cells but not viable cells. In addition, E06 recognizes PC present on microbial cell walls (e.g., *S. pneumoniae*) [17,22]. The V_H_/V_L_ of E06 is identical to that of the prototypic T15 idiotype first described for the IgA NAb, which has been the subject of intense study over many years [23]. Increasing the level of the OSE NAb E06 or the transgenic expression of the single chain variable fragment of the E06 NAb (E06-scFv) provided atheroprotection in *Ldlr^−/−^* mice [17,24,25]. E06 is thought to be atheroprotective by virtue of its ability to prevent the binding of OxLDL to the scavenger receptors CD36 and SR-B1, thereby inhibiting uptake of OxLDL by macrophages [12,26,27] and by neutralizing the pro-inflammatory effects of OxPL [8,12]. Moreover, the E06 IgM NAb has been shown to promote the complement receptor mediated clearance of apoptotic cells by phagocytes in vivo, which may be important for attenuation of atherosclerosis as well as other inflammatory disorders [10].

A wide variety of innate and adaptive immune mechanisms influence the progression of atherosclerosis [28,29]. Natural killer T (NKT) cells are a T cell subset that serve as a bridge between the innate and adaptive immune systems [30]. There are two major classes of NKT cells [30,31,32]. The invariant NKT (iNKT) cells are the most abundant subclass (≥80%) and express a semi-invariant T cell receptor, which in the mouse is Vα14Jα18Vβ8. The non-invariant type II NKT cells express a more diverse T cell receptor repertoire and are present in lower frequency. CD1d is an MHC class I-like molecule that presents lipid antigens, especially glycolipids, and is expressed on several professional antigen presenting cells (macrophages, dendritic cells, B lymphocytes) as well as non-immune cells such as hepatocytes and enterocytes [33,34]. Since both iNKT and type II NKT cells are CD1d restricted, a deficiency of CD1d results in the absence of both classes of NKT cells [35,36]. On the other hand, elimination of the Jα18 chain of the semi-invariant TCR results in a deficiency of only the iNKT cells [37]. 

Despite the recognition of the important protective role of IgM NAbs in health and disease, we are only beginning to understand the regulation of their production. In the course of studies on the role of NKT cells on lipoprotein metabolism and atherosclerosis in *Ldlr^−/−^* mice, we have noted a selective and substantial increase in the plasma titer of E06 IgM in CD1d-deficient *Ldlr^−/−^* (*Cd1d^−/−^Ldlr^−/−^*) mice compared to *Ldlr^−/−^* and Jα18 deficient *Ldlr^−/−^* (*Jα18*^−/−^*Ldlr^−/−^*) mice. The present study was undertaken to gain further understanding of the CD1d-mediated regulation of B-1 cells secreting the E06 NAb that recognizes OSE, including those present on LDL. 

## 2. Materials and Methods

### 2.1. Mouse Strains

*Ldlr^−/−^* mice (stock number 002207) and *Il−10^−/−^* mice (stock number 002251), both on the C57BL/6 background, were originally purchased from Jackson Laboratories and maintained in the vivarium. *Cd1d^−/−^* mice and *Jα18^−/−^* mice, both on the C57BL/6 background, were kindly provided by Drs. Chyung-Ru Wang (Northwestern University) [35] and Albert Bendelac (University of Chicago) [38]. They were crossed with *Ldlr^−/−^* mice and maintained as double knockout mice. Animals were housed in a specific pathogen free facility. *Ldlr^−/−^* mice were used since they have higher levels of plasma LDL, a major source of OSE, than wild type mice. They were maintained on a chow diet and used between 8–15 weeks of age, except in one experiment in which they were fed a Western type diet (Envigo 88137; 21% saturated fat and 0.15% cholesterol) starting at 8 weeks of age. All procedures performed on the mice were in accordance with National Institute of Health guidelines and approved by the Institutional Animal Care and Use Committee at the University of Chicago (ACUP 69271).

### 2.2. Splenic and Peritoneal Cell Preparations

Single-cell suspensions of splenocytes were prepared from spleens and red blood cells were removed by either Ammonium-Chloride-Potassium (ACK) lysis buffer treatment and/or Lympholyte-M (Cedarlane Laboratories Limited, Ontario, Canada) gradient centrifugation. Peritoneal cells were obtained by lavage of peritoneum using 1% bovine serum albumin (BSA) in phosphate buffered saline (PBS). The viability of the cell suspensions and cell numbers were determined with trypan blue. 

### 2.3. Measurement of Antibody Titer by ELISA 

Antibody titers in the plasma of mice were measured using chemiluminescent-based ELISA assays [7,39]. Briefly, the microtiter plates were coated with AB1-2, an IgG T15 anti-idiotype antibody, Cu-OxLDL or MDA-LDL by incubating overnight at 4 °C. After washing, and blocking with 1% BSA/PBS, 50 µL of serially diluted murine plasmas were incubated overnight at 4 °C. The plates were washed with washing buffer, and the captured antibodies were detected using biotinylated antibodies, followed by AP-labeled NeutraAvidin (Pierce Biotechnology Inc. Rockford, IL USA) and a 50% aqueous solution of LumiPhos 530.

### 2.4. ELISPOT Assay 

MultiScreen-IP plates (Millipore, Billerica, MA, USA) were coated with the T15 anti-idiotype IgG AB1-2 (15 µg/mL in PBS), or 4 µg/mL MDA-LDL, Cu-OxLDL, or rat anti-mouse IgM polyclonal antibodies (eBioscience) overnight at 4 °C, followed by blocking with 1% BSA in PBS for 2 h at room temperature. 1–2 × 10^6^ splenocytes or 0.1–0.2 × 10^6^ peritoneal cells per well were seeded into the AB1-2, MDA-LDL and Cu-OxLDL coated wells in a final volume of 100 µL RPMI 1640 medium supplemented with 10% FCS, 100 units/mL penicillin G, and 100 µg/mL streptomycin. For IgM coated wells, one tenth the number of splenocytes or peritoneal cells were seeded into the wells. Peritoneal B-1 cells were incubated with 5 µg/mL LPS (E. coli serotype O55-B5, Sigma-Aldrich, St. Louis, MO, USA) to stimulate immunoglobulin secretion. Cells were incubated at 37 °C in 5% CO_2_ for 24 to 36 h, washed with PBS-0.05% Tween-20, then treated with biotinylated anti-IgM antibody (BD Biosciences, San Jose, CA, USA) followed by incubation of peroxidase-conjugated streptavidin (BD Biosciences) according to the manufacture’s protocol. The spots were visualized using TrueBlue peroxidase substrate (KPL, Gaithersburg, MA, USA) and analyzed on an Immunospot Analyzer (Cellular Technology, Inc, Cleveland, OH, USA). In parallel, an ELISPOT assay was carried out with an uncoated plate to determine background spot staining (typically 0–2 spots), which was then subtracted from the values from the antigen specific spots. 

### 2.5. Flow Cytometry and Cell Sorting

Single cell suspensions of splenocytes or peritoneal cells in FACS buffer (1% BSA in PBS) were prepared and then stained with fluorochrome labeled antibodies against mouse PerCP-Cy5.5 or FITC anti-CD19 (1D3), FITC anti-CD21 (4E3), PE anti-CD23 (B3B4), PerCP-Cy5.5 anti-CD5 (53–7.3), and APC anti-IL-10 (JESt-16E3) purchased from eBioscience. Prior to intracellular staining for IL-10, the cells were treated with phorbol 12-myristate 13-acetate (PMA), ionomycin and protein transport inhibitor (eBioscience, San Diego, CA, USA). Samples were analyzed on a FACSCanto flow cytometer (BD Bioscience). Splenic B lymphocytes subsets were sorted with a FACSAria cell sorter (Becton Dickinson) after staining with PE-anti-CD23, PerCP-Cy5.5-CD19, and FITC-anti-CD21 antibodies. The sorted cells were analyzed for E06 and total IgM-secreting cells (ISC) by ELISPOT as described above.2.6. Statistical Analysis

Data are presented as mean ± standard error of the mean (SEM). When comparing more than two groups, the data were analyzed by one-way ANOVA followed by Bonferroni post-hoc test using StatView version 5.0.1. When comparing two groups, the data were analyzed by Student’s *t* test. Significance was set at *p* < 0.05.

## 3. Results

### 3.1. Increased Titer of E06 NAb in Cd1d^−/−^Ldlr^−/−^ Mice

The plasma titers of the E06 NAb in Western type diet fed female *Cd1d*^−/−^*Ldlr*^−/−^ mice is more than 20-fold higher compared to *J*α*18^−/−^Ldlr*^−/−^ and *Ldlr*^−/−^ mice and 10-fold higher in male mice (Figure 1A). This increase is specific for the E06 IgM NAb since there were no corresponding fold changes in total IgM levels. in IgM titers to copper oxidized LDL (Cu-OxLDL), or to malondialdehyde-modified LDL (MDA-LDL), a dominant target of B-1 cell antibodies [7] (Figure 1B). Titers of IgG antibodies recognizing MDA-LDL and Cu-OxLDL, and the level of the oxidation epitope on apoB-containing lipoproteins recognized by E06 were not different between the three groups (data not shown). Increased levels of E06 IgM without increases in total IgM levels were also observed in chow-fed female *Cd1d*^−/−^*Ldlr*^−/−^ mice (4.5-fold higher, *p* < 0.001) and *Cd1d^−/−^* mice expressing the LDLR (1.8-fold higher, *p* < 0.02). Since the chow fed animals have significantly lower plasma lipid levels than high fat diet fed mice, this suggests that though hyperlipidemia may exacerbate the effect it is not a requirement. All remaining studies were performed on chow-fed *Ldlr^−/−^* mice. These results suggest that CD1d may directly or indirectly suppress the production of the E06 NAb in a highly selective fashion. Further, since the plasma titers of E06 IgM were increased in *Cd1d^−/−^Ldlr^−/−^* mice but not in *J*α*18^−/−^Ldlr*^−/−^ mice, which both lack iNKT cells, this suggests that CD1d expressing cells or CD1d-restricted cells, but not iNKT cells, are likely involved in the CD1d-mediated inhibition of E06 IgM expression. 

### 3.2. Increased E06 IgM-Secreting B-1 Cells in the Spleens of Cd1d^−/−^Ldlr^−/−^ Mice

ELISPOT assays were performed to investigate whether the high plasma titer of E06 IgM in *Cd1d^−/−^Ldlr^−/−^* mice is due to an increased number of B-1 cells expressing the antibody. Both total and E06 ISC in the peritoneum and spleen were quantitated. E06 ISC in the spleens of chow fed *Cd1d^−/−^Ldlr^−/−^* mice were >5-fold higher in female and 2-fold higher in male mice compared to *Ldlr^−/−^* mice (Figure 2A and Appendix A). This is consistent with the higher plasma levels noted in females vs. males and suggests that the CD1d-mediated regulation of E06 ISC is partly influenced by gender. Consistent with the lack of a role of iNKT cells, E06 ISC in the spleen of *Jα18^−/−^Ldlr^−/−^* mice are comparable to that in *Ldlr^−/−^* mice (Figure 2A). The total number of splenic ISC was not influenced by the absence of CD1d, indicating that the increased level of E06 ISC in *Cd1d^−/−^Ldlr^−/−^* mice was not due to a general increase in IgM-secreting B cells. This is consistent with equal titers of total IgM in plasma (Figure 1) and also with the equal number of CD19^+^IgM^+^ cells in the spleens of the groups of mice as examined by flow cytometry (data not shown). The total numbers of splenocytes in *Ldlr^−/−^*, *Jα18^−/−^Ldlr^−/−^* and *Cd1d^−/−^Ldlr^−/−^* mice were also similar. Furthermore, the number of splenic cells secreting anti-MDA-LDL IgM (Figure 2A) or anti-Cu-OxLDL IgM (data not shown) was not altered in the absence of CD1d. 

In contrast to the situation in the spleen, peritoneal E06 ISC did not significantly vary as a function of CD1d expression level (Figure 2B), indicating that the spleen is the likely site of regulation of E06 IgM expression by CD1d. Interestingly, males had more E06 IgM-secreting B-1 cells in the peritoneum than females in both *Ldlr^−/−^* and *Cd1d*^−/−^*Ldlr*^−/−^ mice (*p* = 0.01 and 0.0005, respectively). The plasma titers of E06 NAb and the number of E06 ISC in the spleen showed the same female bias, in contrast to the male bias in ISC in the peritoneal cavity, further indicating that the spleen, not the peritoneal cavity, is the likely site of CD1d-mediated regulation of E06 IgM expression.

E06 mRNA level was quantitated in the spleens of *Ldlr^−/−^* and *Cd1d*^−/−^*Ldlr*^−/−^ mice using primers specific for the E06 V_H_ CDR3 region. The trend was for higher E06 mRNA levels in *Cd1d*^−/−^*Ldlr*^−/−^ mice compared to *Ldlr^−/−^* mice, but the results were not significantly different (Appendix A). 

### 3.3. Total B-1 Cells Are Not Increased in the Spleen of Cd1d^−/−^Ldlr^−/−^ Mice

The spleens of female *Cd1d*^−/−^*Ldlr*^−/−^, *Ja18^−/−^Ldlr^−/−^* and *Ldlr*^−/−^ mice were analyzed by flow cytometry to determine if there were major differences in B lymphocyte populations. The B lymphocytes were separated into CD19^+^CD23^hi^CD21^int^ (follicular (FO) B cells), CD19^+^CD23^lo^CD21^−^ (B-1 cells) and CD19^+^CD23^lo^CD21^hi^ (marginal zone (MZ) B cells) (Figure 3A). Based on the percent of cells in each population and the total splenocyte numbers, no difference in the number of MZ B cells and CD19^+^CD23^lo^CD21^−^ B cells was observed, but the number of FO B cells was 21% lower in the spleens of *Cd1d^−/−^Ldlr^−/−^* mice (Figure 3B). 

E06 IgM has previously been demonstrated to be secreted exclusively by B-1 cells [20]. To ascertain whether a non-B-1 cell population was secreting E06 IgM NAb in the context of CD1d deficiency, ELISPOT assays were performed on the FO B cells, B-1 cells and MZ B cells isolated by FACS from the spleens of *Cd1d*^−/−^*Ldlr*^−/−^ mice. The results confirm that the E06 ISC in the *Cd1d*^−/−^*Ldlr*^−/−^ mice were only found in the CD19^+^CD23^lo^CD21^−^ population containing B-1 cells (Figure 3C). On the other hand, as expected, IgM^+^ cells were also detected in the MZ B cell populations. 

### 3.4. Reduced Number of Splenic CD19^+^CD5^+^IL-10^+^ Cells in Cd1d^−/−^Ldlr^−/−^ Mice

The expanded number of splenic E06 ISC in the absence of CD1d suggests that E06 ISC are suppressed directly or indirectly by CD1d expression. One possible explanation is that regulatory cells (i.e., regulatory B and/or T cells) [40,41] are present in the spleen of wild type mice and that they regulate the number of E06 ISC. The number of those putative suppressor cells might be reduced or their activity might become functionally impaired in the absence of CD1d. B regulatory cells are defined functionally by their production of regulatory cytokines such as IL-10 and TGFβ and have been shown to suppress immune responses in autoimmunity models [40]. IL-10 B cells are IL-10 secreting CD5^+^ B lymphocytes that express high levels of surface CD1d [42,43]. IL-10-producing CD19^+^ CD5^+^ B cells are reduced in the spleen of *Cd1d*^−/−^*Ldlr*^−/−^ mice by >50% relative to *Ldlr^−/−^* mice, but not in the peritoneal cavity (Figure 4A) suggesting that the microenvironment may influence the induction of these potential regulatory cells. The frequency of IL-10 producing CD19^+^ B cells in the peritoneum was much higher than in the spleen, which is consistent with a previous finding that there is more IL-10 mRNA present in the peritoneal B-1 population than in splenic B-1 or B-2 cells [44]. MZB cells, a subset of B-2 cells, are also dominant IL-10-producing cells in certain contexts [45]. They represent a first-line of host defense against pathogen infection similar to B-1 cells and infection-induced antibody production by MZB cells is influenced by CD1d [46]. However, neither the number (Figure 4B) nor frequency (data not shown) of MZB IL-10^+^ cells was significantly different in *Cd1d*^−/−^*Ldlr*^−/−^ mice. 

T_H_ and T_reg_ cells also play important roles in immune regulation, mainly through cytokine secretion [41]. However, there was no significant difference in total number of splenic CD3^+^ T cells, the number of IL-10^+^, IL-4^+^, IFNγ^+^ or IL-5^+^ splenic T cells, or the number of T_reg_ and IL-10^+^ producing T_reg_ populations between these three strains of mice (Appendix A). 

### 3.5. E06 ISC in the Spleen of Il10^−/−^ Mice

The flow cytometry analysis described above demonstrated that IL-10-producing CD19^+^CD5^+^ B cells in the spleen were reduced in *Cd1d*^−/−^*Ldlr*^−/−^ mice. In general, IL-10 has immune suppressive functions, and IL-10 produced by B-1 cells has been shown to specifically inhibit the proliferation of B-1 cells in an autocrine fashion [47]. Thus, diminished IL-10 mediated suppressive regulation of the number of the E06 IgM-secreting cells might be a possible explanation for the increase in E06 IgM-secreting cells in the spleen of *Cd1d* deficient mice. To gain insight into whether IL-10 might regulate E06 expression, an ELISPOT was performed on splenocytes from *Il-10^−/−^* mice. The number of E06 ISC in the spleen of *Il-10^−/−^* mice was significantly higher than in C57BL/6 mice (Figure 5). This increase was specific as the number of MDA-LDL ISC was not significantly increased in *Il-10^−/−^* mice. Although there was a trend towards an increased number of total IgM-secreting cells in the spleen, the difference was not significant. If anything, this would lower the difference in the number of E06 ISC in the spleen of *Il-10^−/−^* mice since the data are expressed per IgM expressing cells. 

## 4. Discussion

In this study, we observed a substantial increase in the plasma titers of the E06 IgM NAb that recognizes the PC of OxPL in *Cd1d*^−/−^*Ldlr*^−/−^ mice. This increase appears to be due primarily to an increase in the number of E06 ISC in the spleen but not in the peritoneal cavity. The increase in both E06 IgM titers and E06 ISCs secreting the NAb is highly selective, as total ISC levels are not increased and neither the titer of total IgM or IgM directed against MDA-epitopes, another OSE found on OxLDL and apoptotic cells, are increased. CD1d is a restriction element for both iNKT and type II NKT cells. iNKT cells apparently do not participate in this selective regulation as we found no difference in E06 NAb titers or number of E06 ISC in *Jα18^−/−^Ldlr*^−/−^ mice that lack iNKT cells. This is in contrast to the reported autoreactive CD1d^+^ B cell, whose activation is limited by iNKT cells [48]. The most likely explanation for our observation is that a CD1d-dependent process not involving iNKT cells suppresses the number of E06 ISC or the expression of E06 IgM in B-1 cells. Immune cells that may be involved in this suppression include type II NKT cells and regulatory B cells. In addition, Vδ1^+^ γδ T cells may also have a role as this subset of cells has been shown to recognize lipids presented by CD1d, at least in humans [49,50]. However, the involvement of CD1d expressing dendritic cells, macrophages and epithelial cells cannot be ignored. 

The high plasma level of E06 NAb in *Cd1d*-deficient mice appears to be due primarily to the expansion of E06 ISC in the spleen. This is consistent with splenic B-1 cells being more dominant producers of IgM than peritoneal B-1 cells [51]. This implies that the CD1d-dependent regulatory cells suppressing E06 ISC are primarily functioning in the spleen. Suppressive regulatory cells could be reduced or activating regulatory cells increased in the absence of CD1d. Alternatively, CD1d may limit the migration of B-1 cells from the peritoneal cavity to the spleen, where they can divide and differentiate [52]. There is evidence that the homing or retention of B-1 cells in various locations may be differentially regulated [53,54]. For example, the chemokine CXCL13 influences the homing or retention of some B-1 cells in the peritoneum and omentum, but not in the spleen [55]. Interestingly, *Cxcl**13* deficient mice have reduced plasma levels of anti-PC IgMs without any differences in total IgM levels. The enhanced E06 IgM titers in the CD1d deficient mice, whatever mechanism is involved, must be relatively specific for the E06 ISC since neither total splenic B-1 cells nor B-1 cells producing NAbs against other OSE on LDL are altered by the absence of CD1d. 

CD1d-mediated regulation is likely independent of IL-5, a cytokine known to increase E06 titers [39], since the numbers of IL-5 producing T cells in the spleen of *Cd1d^−/−^Ldlr^−/−^* mice were not altered. On the other hand, splenic CD19^+^ CD5^+^ IL10^+^ cells are reduced in *Cd1d*^−/−^*Ldlr*^−/−^ mice (Figure 4) as are splenic levels of IL-10 mRNA (data not shown). Consistent with this, E06 ISC are increased in the spleen of *Il-10^−/−^* mice, suggesting that at least part of the mechanism by which CD1d regulates E06 ISC may involve IL-10 producing cells. IL-10 is secreted by several cell types, including regulatory IL-10 B cells that express high levels of CD1d [42]. Studies in intestinal epithelial cells have shown that ligation of CD1d leads to increased expression of IL-10 [56,57]. However, if IL-10 is involved it appears that the secretion of IL-10 by specific subsets of splenic cells may be more critical, since comparable levels of splenic IL-10 producing MZB cells, T cells and T_reg_ cells were observed in *Ldlr^−/−^* and *Cd1d*^−/−^*Ldlr*^−/−^ mice. Even if IL-10 is a participant in the suppression of the production of antibodies by splenic B-1 cells, our data do not address why this is specific for B-1 cells secreting E06 IgM among other IgM-secreting B-1 cells. 

It is worth noting that splenic MZ B cells, which have many phenotypic and functional characteristics of B-1 cells [58], do not appear to be operative in the regulation of E06 ISC. While they express high levels of CD1d, their numbers were not influenced by the absence of CD1d (Figure 3). However, it has been reported that infection-induced activation of MZB cells, as well as their pathogen-specific IgM production depend on the expression of CD1d [46]. This opposite effect of CD1d on the activation of IgM producing MZB on one hand, and the suppression of E06 IgM production by B-1 cells on the other hand is a novel mechanism in regulating two functionally similar but different subsets of innate-like immune cells. 

The global absence of *Cd1d* in mouse models of atherosclerosis is associated with a reduction in atherosclerosis [34]. This may be due to decreased secretion of proinflammatory cytokines and/or cytotoxic proteins that promote apoptosis of cells in atherosclerotic lesions. The increased titer of atheroprotective E06 NAbs in *Cd1d^−/−^* mice may also have contributed to the decreased atherosclerosis. Since these antibodies are even increased in chow fed mice, this perhaps reflects a more profound role in immune homeostasis, even in the absence of hypercholesterolemia. CD1d is expressed on several professional antigen presenting cells as well as non-immune cells such as hepatocytes and enterocytes. It presents hydrophobic lipid antigens, especially glycolipids, to NKT cells [30,31,32,33], which are important in immuno-regulation, including early cytokine responses and promoting B cell antibody production [59,60]. Our finding with *Jα18^−/−^ Ldlr^−/−^* mice suggests that iNKT cells are not involved in E06 regulation. On the other hand, we could not preclude autocrine-dependent signaling influencing E06 IgM regulation initiated by the cytoplasmic tail of CD1d on B cells as has been found upon activation of intestinal epithelial cells by CD1d-ligation [56]. 

The E06 class of innate IgM have been shown to play multiple roles in innate responses to both pathogen and non-pathogen danger-associated molecular patterns [9]. This fine tuning of E06 IgM expression might be physiologically important to critically adjust the generation of innate E06 IgM autoantibody to facilitate inactivation of pathogens, apoptotic cells and oxidized lipids as needed but to avoid harmful autoimmunity. There is clearly a great deal of work that remains to be undertaken to understand these processes and their significance. The elucidation of the nature of the regulation of E06 ISC could further our understanding of the importance of innate immunity against pathogens as well as homeostatic housekeeping functions in health and disease. 

## Figures and Tables

**Figure 1 antibodies-09-00030-f001:**
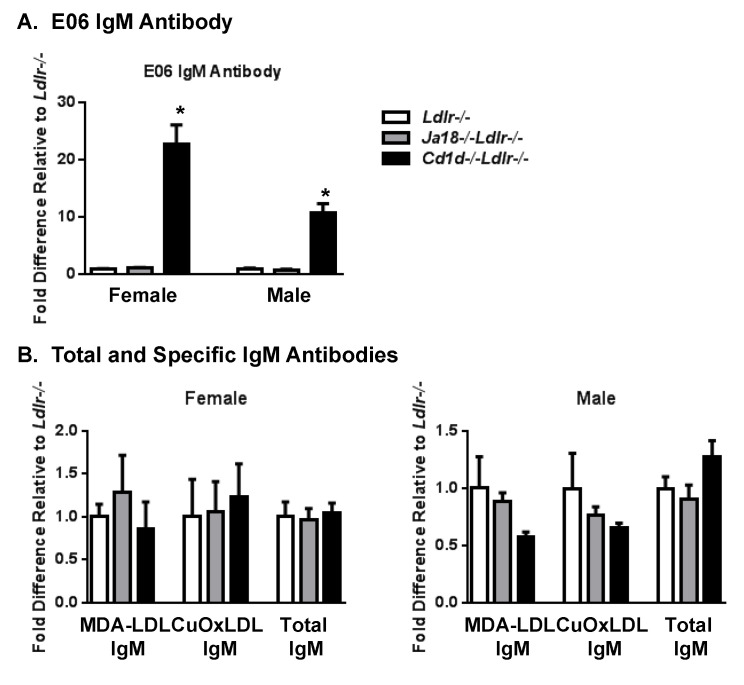
Selective increase in E06 IgM antibody in the plasma of *Cd1d*^−/−^*Ldlr*^−/−^ mice. Relative plasma IgM titers specific for (**A**) E06 IgM (1:100 dilution), and (**B**) MDA-LDL (1:200 dilution), Cu-OxLDL (1:200 dilution) and total IgM (1:750 dilution) in female and male *Ldlr^−/−^*, *Jα18^−/−^Ldlr^−/−^* and *Cd1d^−/−^Ldlr^−/−^* mice fed western type diet for 4 weeks were determined by ELISA. The level of each antibody is expressed relative to the level in *Ldlr^−/−^* mice. The results are expressed as the mean ± SEM with 4 to 8 animals per group. Significance: * *p* < 0.0001 vs. *Ldlr^−/−^* mice.

**Figure 2 antibodies-09-00030-f002:**
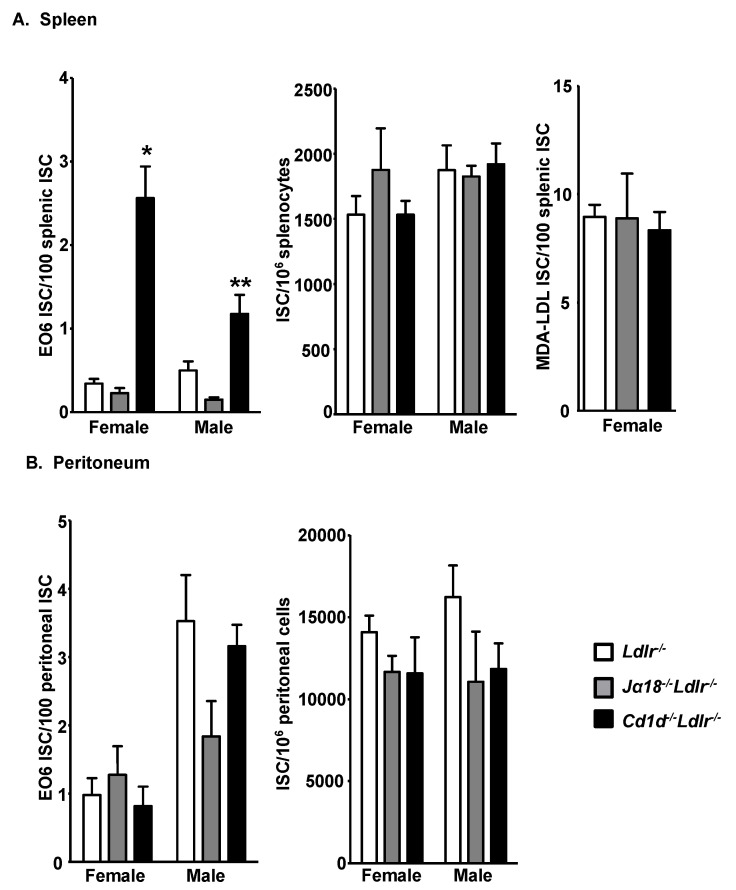
E06 IgM-secreting cells (ISC) are increased in the spleen of *Cd1d*^−/−^*Ldlr*^−/−^ mice. Quantitation of the ELISPOT analysis of cells in the (**A**) spleen and (**B**) peritoneal cavity of chow fed *Ldlr^−/−^*, *Jα18^−/−^Ldlr^−/−^* and *Cd1d^−/−^Ldlr^−/−^* mice producing E06 IgM, total IgM and MDA-LDL IgM. The number of E06 and MDA-LDL IgM-secreting cells are expressed per 100 ISC. The results are the mean ± SEM of *n* = 11–14 *Ldlr^−/−^* and *Cd1d^−/−^Ldlr^−/−^* mice pooled from triplicate independent experiments and *n* = 3–7 *Jα18^−/−^Ldlr^−/−^* mice pooled from two independent experiments for splenocytes and *n* = 5–14 *Ldlr^−/−^* and *Cd1d^−/−^Ldlr^−/−^* mice pooled from two independent experiments and *n* = 3–5 *Jα18^−/−^Ldlr^−/−^* mice pooled from two independent experiments for peritoneal cells. Significance: * *p* < 0.0001 and ** *p* < 0.01 vs. *Ldlr^−/−^* mice.

**Figure 3 antibodies-09-00030-f003:**
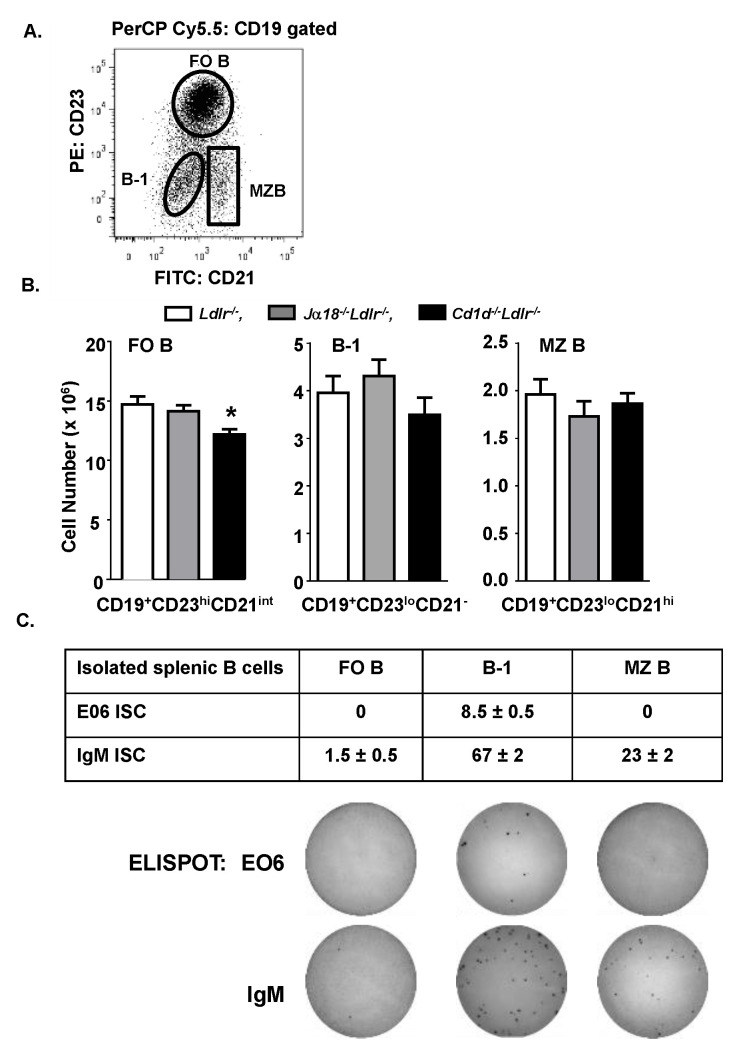
B-1 cells are not increased in the spleen of *Cd1d*^−/−^*Ldlr*^−/−^ mice. (**A**) Representative flow cytometric analysis of splenic B cells subsets. (**B**) Based on the percent of cells in the indicated populations and the total number of splenocytes, the number of CD19^+^CD23^hi^CD21^int^ (FO B cells), CD19^+^CD23^lo^CD21^−^ cell (B-1 cells), and CD19^+^CD23^lo^CD21^hi^ (MZ B cells) in the spleens of chow-fed female *Ldlr^−/−^*, *Jα18^−/−^Ldlr^−/−^* and *Cd1d*^−/−^*Ldlr*^−/−^ mice were quantitated. The results are the mean ± SEM of *n* = 13–14 pooled from four independent experiments. Significance: * *p* = 0.002 vs. *Ldlr^−/−^* mice. (**C**) Splenic B cell populations in *Cd1d^−/−^Ldlr^−/−^* mice isolated by FACS as outlined in (**A**) were analyzed for E06 ISC or total ISC by ELISPOT. The results are the mean ± SEM of one experiment with cells isolated from two mice analyzed in duplicate.

**Figure 4 antibodies-09-00030-f004:**
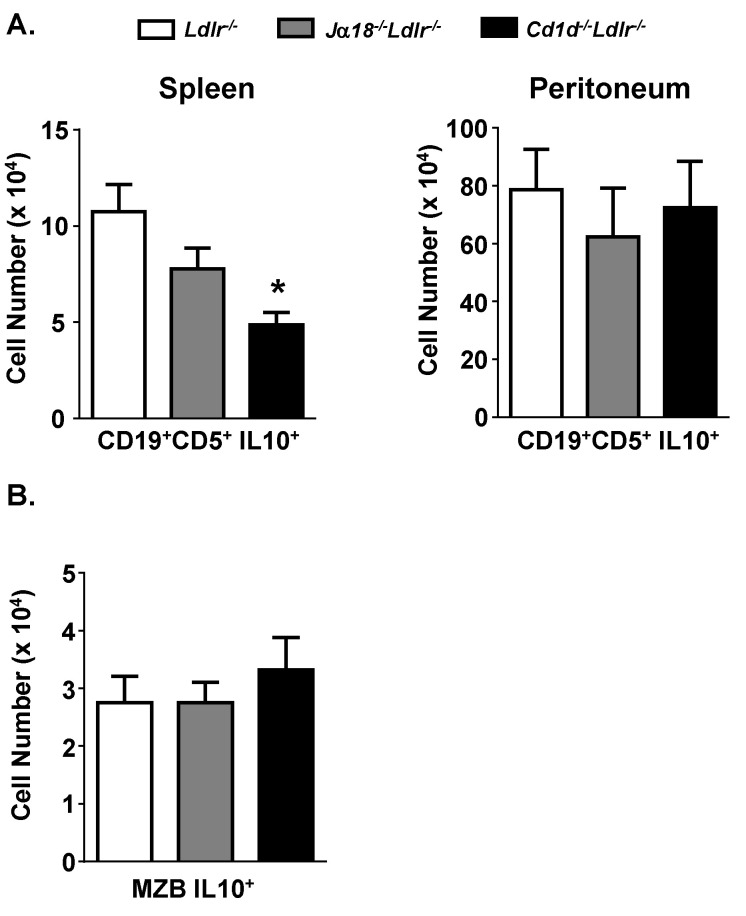
(**A**) Reduced CD19^+^ CD5^+^ IL10^+^ B lymphocytes in the spleen of *Cd1d*^−/−^*Ldlr*^−/−^ mice. Splenocytes and peritoneal cells from chow fed female mice were analyzed for CD19^+^CD5^+^ IL-10^+^ B cells via flow cytometry. (**B**) Splenocytes were also analyzed for IL10^+^ MZB cells (CD19^+^CD23^lo^CD21^hi^) by flow cytometry. The results are the mean ± SEM of *n* = 7–9 pooled from two independent experiments. Significance: * *p*= 0.001 vs. *Ldlr^−/−^* mice.

**Figure 5 antibodies-09-00030-f005:**
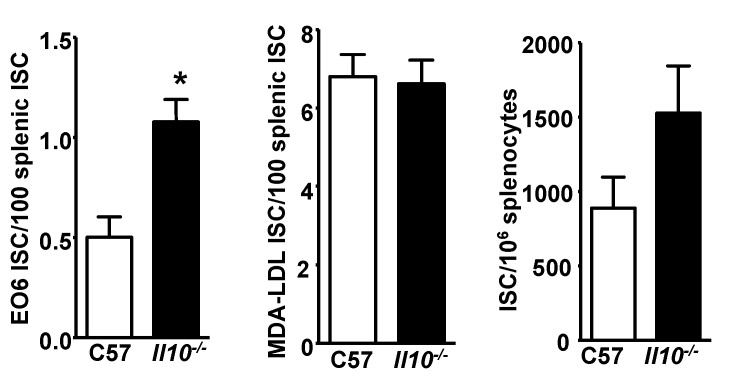
Increased levels of E06 ISC in the spleen of *Il-10^−/−^* mice. E06 ISC, MDA-LDL ISC and total ISC in the spleen of 8-week old chow fed female C57BL/6 and *Il10^−/−^* mice were identified by ELISPOT assay. The results are the mean ± SEM of *n* = 13–15 pooled from three independent experiments. Significance: * *p* < 0.001.

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
