# Peer review of "CD1d Selectively Down Regulates the Expression of the Oxidized Phospholipid-Specific E06 IgM Natural Antibody in Ldlr−/− Mice"

_2073-4468, 2020, doi:10.3390/antib9030030_

Round 1
Reviewer 1 Report
The study by Biswas and colleagues show that titers of the natural antibody E06 which recognises oxidised forms of the phospholipid phosphatidylcholine are significantly increased in CD1d knock out mice. The results indicate that through a CD1d mediated mechanism, there is an expansion of the E06 producing B-1 B cells specifically within the spleen. Furthermore, this cd1d-mediated regulatory network seems to involve the suppressive cytokine IL-10 and the IL-10 producing CD19+ CD5+ B cells. The data is of great interest, as it provides evidence that the CD1d protein may regulate innate immune responses independently of its well-known role in activating NKT cells, hence the current study indicates a previously unknown function for CD1d in regulating immunity. The results are clear and are well presented. I have several comments
- The authors show an increase in the E06 ISC in IL10 knock out mice, did they measure the E06 titres in these mice?
- In the IL-10 knockout mice, were the E06 ISC only increased in the spleen?
- There is evidence that CD1d regulates responses via its cytoplasmic tail, could this be contributing to the regulation of B-1 cells? The authors mention this as a possible mechanism in their discussion. Did the authors investigate the E06 producing B-1 cells in the wild type and CD1d knock out mice, and whether sorted populations produced more or less antibody in the absence of cross-talk with other immunocytes?
- I suppose following on from the previous comment, do the CD1d knock-out E06 ISC produce more antibody or is the change in titer just due to a greater of expansion of these ISC? This would further support the conclusion that this regulation is driven by cellular cross talk
Author Response
Thank you for your thoughtful review of the manuscript.
- The authors show an increase in the E06 ISC in IL10 knock out mice, did they measure the E06 titres in these mice?
Response: No, we did not measure E06 titers in the plasma of the IL-10 mice. Unfortunately, the samples are no longer available.
- In the IL-10 knockout mice, were the E06 ISC only increased in the spleen?
Response: Prior to investigating IL-10 our studies demonstrated an effect of CD1d deficiency on E06 ISC only in the spleen. Therefore. we did not measure E06 ISC in the peritoneal cavity of the IL-10 deficient mice.
- There is evidence that CD1d regulates responses via its cytoplasmic tail, could this be contributing to the regulation of B-1 cells? The authors mention this as a possible mechanism in their discussion. Did the authors investigate the E06 producing B-1 cells in the wild type and CD1d knock out mice, and whether sorted populations produced more or less antibody in the absence of cross-talk with other immunocytes?
Response: Cross-talk with other immune cells is certainly a possible mechanism by which CD1d may regulate E06 antibody production. Our demonstration of decreased levels of CD5+ IL-10+ B cells in the spleen in the absence of CD1d and that IL-10 deficiency increases E06 ISC suggests that IL-10 B cells are a candidate immune cells that regulates the number of this subset of B1 cells in the spleen. We did not, however, explore this further, in part because we are not able to specifically isolate the E06 expressing B1 cells for coculture experiments.
- I suppose following on from the previous comment, do the CD1d knock-out E06 ISC produce more antibody or is the change in titer just due to a greater of expansion of these ISC? This would further support the conclusion that this regulation is driven by cellular cross talk
Response: We did consider that the antibody secretion by the E06 ISC might be increased in the absence of CD1d. However, we could not assess this due to the inability to specifically isolate the E06 expressing B1 cells by FAC sorting. On the other hand, the 2 to 5-fold increase in total number of E05 ISC in the spleen is likely to significantly contribute to plasma titers of the E06 antibody in the plasma.

Reviewer 2 Report
General remarks:
Even though the title of the paper indicates the critical importance of CD1d to the expression of the natural antibody, none of the experiments directly addressed this by using CD1d-/- mice. Instead, the double negative Cd1d-/-Ldlr-/- mice were analyzed. This rationale of this choice of animals has not been explained.
The key to the analysis of various subsets of B cells is the flow cytometry. Although antibodies against CD19 (1D3), CD21 (4E3), CD23 (B3B4), B220 (RA3-6B2), CD43 (eBioR2/60), CD5 (53-7.3), IL10 (JESt-16E3) and IgM (eB121-15F9) were listed, the fluorochromes conjugated with each of these antibodies were not indicated. This is absolutely critical to assess how the analysis was carried out. In Fig. 3A and B, CD19, CD23 and CD21 antibodies were used. In Fig. 3C, CD19, IgM, CD23, CD43, and CD5 were sequentially analyzed. It should be made clear how spectral overlaps between the fluorochromes used influenced the results.
Fig. 3C-The gating strategies were not clearly explained. The arrows should be properly placed on the gated population of interest. For instance, the arrow should be placed on the CD19-gate. What does the cursor positioned on the IgM+ cells mean? It appears to be arbitrarily placed and does not enclose all of the IgM+ cells, why? Again, presumably CD43-gated cells are further analyzed for CD5 but was not properly indicated. Presumably, CD23+ and CD43+ cells were analyzed on the ‘dim IgM+’ gated cells, why? Please explain how will it affect the results?
Fig. 4. Please clarify whether IL-10 was detected on the cell surface or by intracellular staining with or without stimulation?
Fig. 5. The numbers of EO6+ cells are too low to make a conclusion.
Discussion:
It was repeatedly argued that CD1d deficiency enhanced E06 IgM antibodies. The significance of analyzing Ldlr-/- mice was not explained. The critical question is whether CD1d deficiency directly influenced the natural antibody production remained unanswered.
Minor points:
Introduction is complex and long and could be simplified.
Statistics: Both the t-test and the p values should be italicized.
Author Response
Thank you for your thoughtful review of the manuscript. We have made changes based on your comments that we think improve the manuscript.
- The key to the analysis of various subsets of B cells is the flow cytometry. Although antibodies against CD19 (1D3), CD21 (4E3), CD23 (B3B4), B220 (RA3-6B2), CD43 (eBioR2/60), CD5 (53-7.3), IL10 (JESt-16E3) and IgM (eB121-15F9) were listed, the fluorochromes conjugated with each of these antibodies were not indicated. This is absolutely critical to assess how the analysis was carried out. In Fig. 3A and B, CD19, CD23 and CD21 antibodies were used. In Fig. 3C, CD19, IgM, CD23, CD43, and CD5 were sequentially analyzed. It should be made clear how spectral overlaps between the fluorochromes used influenced the results.
Response: The fluorochromes used for the flow cytometry in Figure 3A has been added to Materials and Methods and the Figure legend. We decided to eliminate figure 3C. Figure 3A demonstrates that the total number of B-1 cells are not different. Figure 3C provides data on the number of B1a and B1b subsets in the spleen, but the subsets are not the focus of the study and the cells in the CD19hi IgMhi, CD23-CD43+ gate are very low.
- 3C-The gating strategies were not clearly explained. The arrows should be properly placed on the gated population of interest. For instance, the arrow should be placed on the CD19-gate. What does the cursor positioned on the IgM+ cells mean? It appears to be arbitrarily placed and does not enclose all of the IgM+ cells, why? Again, presumably CD43-gated cells are further analyzed for CD5 but was not properly indicated. Presumably, CD23+ and CD43+ cells were analyzed on the ‘dim IgM+’ gated cells, why? Please explain how will it affect the results?
Response: As mentioned above we decided to eliminate figure 3C from the manuscript.
- 4. Please clarify whether IL-10 was detected on the cell surface or by intracellular staining with or without stimulation?
Response: We apologize for not including this information in the initial submission. IL-10 was detected by intracellular staining following stimulation of the cells with PMA and ionomycin in the presence of a protein transport inhibitor containing Brefeldin A and monensin. This information was added to the Materials and Methods and the legend was revised to indicate that the intracellular IL-10 was being detected.
- 5. The numbers of EO6+ cells are too low to make a conclusion.
Response: We agree that the number of E06 ISC is low. We were limited by the number of cells that we obtained after FAC sorting. We have shown previously that B1 cells produce E06 (reference 20 in the manuscript). While we would not attempt to make a quantitative assessment of the fold increase in the production of E06 and IgM by these B cell subsets, the results do suggest that it is the B1 cell fraction not the FO B cells or MZ B cells that produce E06.
Discussion:
- It was repeatedly argued that CD1d deficiency enhanced E06 IgM antibodies. The significance of analyzing Ldlr-/- mice was not explained. The critical question is whether CD1d deficiency directly influenced the natural antibody production remained unanswered.
Response: We apologize for not making this clear in the initial submission. Our laboratories are interested in the atherogenic role of natural killer T cells and as well as E06 and other natural antibodies that recognize oxidation specific epitopes. We used Ldlr-/- mice for the current studies to increase plasma LDL levels, a major source of oxidation specific epitopes since, unlike humans, mice have very low levels of LDL. This information has been added to Materials and Methods where we describe the mouse models and in the introduction.
The reviewer expresses concern about the performance of our experiments in the context of LDLR deficiency. This is complex question as the absence of the receptor could have an effect on the level of plasma lipids and a lipid independent effect on cell behavior. We should point out that except in the experiments reported in figure 1 chow diet was fed to the animals. On chow the increase in plasma cholesterol is very modest[give example]. However, whatever the effect of these potential perturbations, they are quite selective for spleen and not peritoneal B-1 cells as illustrated in figure 2.
Minor points:
- Introduction is complex and long and could be simplified.
Response: The introduction has been shortened by simplifying the discussion of the role of E06 in atherosclerosis and other disorders.
- Statistics: Both the t-test and the p values should be italicized.
Response: Changed as requested.

Round 2
Reviewer 2 Report
The authors have decided to delete Fig. 3C but retained the data that were originally depicted as Fig. 3D (now Fig. 3C). In that case, it should be clearly explained how did they arrive at the numbers of total FO B, B-1 and MZ B in the text.
Author Response
To clarify aspects of how the studies in Figure 3 were performed the following changes were made:
- We added a new sentence (lines 137- 138) to indicate that the FACS sorted cells were analyzed by ELISPOT.
- Lines 212-215 and the legend to Figure 3 were revised to indicate that the number of each of the B cell populations were derived from the percent of cells in each gate and the total number of splenocytes.
- Lines 218-219 were slightly revised for clarity.